# Safety Evaluation of Subway Tunnel Construction under Extreme Rainfall Weather Conditions Based on Combination Weighting–Set Pair Analysis Model

**Ping Liu, Yu Wang, Tongze Han \*, Jiaming Xu and Qiangnian Li**

School of Civil Engineering, Lanzhou University of Technology, Lanzhou 730050, China
\* Correspondence: hantzacademic@163.com

**Abstract:** Regional extreme rainfall events have occurred frequently in China, and subway tunnel construction faces possible threats under extreme weather conditions. Thus, in this study, we used the set pair analysis (SPA) approach to the construction safety evaluation of subway tunnels and developed a construction safety evaluation model under extreme rainfall circumstances. Firstly, based on careful consideration of the complex construction environment of subway tunnels under extreme rainfall weather conditions, a construction safety evaluation system of subway tunnels was developed considering four aspects: rainfall, hydrogeology, construction design, and management. Moreover, the weighting analysis of each index factor was carried out using the improved analytic hierarchy process (IAHP) method, the entropy weight method (EWM), and the linear weighting method. Secondly, considering the uncertainty of subway tunnels' construction safety evaluation system and the fuzzy nature of evaluation-level classification, a construction safety evaluation system of subway tunnels based on the multivariate linkage number and set pair analysis theory was established. Finally, we applied the model to a subway tunnel construction case. The results show that the evaluation results are consistent with the actual engineering survey results, which verifies the practicality and effectiveness of the model in evaluating subway tunnel safety. We also determined the primary factors and risk development trends that affect the safety of subway tunnel construction under extreme rainfall weather conditions to guide the safety risk management of subway tunnel construction.

**Keywords:** subway tunnel construction; extreme rainfall weather; construction safety assessment; set pair analysis

## 1. Introduction

The 21st century is the era of significant tunnelling and underground engineering construction developments. As a major tunnel and underground engineering construction market, urban rail transit construction in China has entered an unprecedented boom period. By the end of 2020, there were 57 cities with urban rail transit under construction, 32 cities with transit completed and opened to traffic, and 182 lines in operation [1]. Moreover, with the continuous expansion of the scale of the urban rail transit line network, cities such as Beijing and Shanghai are still promoting the construction of the subway with annual investments of tens of billions of yuan. However, in recent years, as the rate of global warming has continued to increase, which has intensified the climate system's instability and led to changes in the distribution and intensity of the weather system, regional extreme rainfall weather events have frequently and unexpectedly occurred. Subway tunnel construction accidents induced by rainfall infiltration have frequently occurred throughout the country, bringing severe threats to the safe construction of subways [2]. According to the assessment report "Climate Change 2021: The Natural Science Basis", in the coming decades, extreme weather events will become more frequent as the warming of the global climate continues and becomes the "new normal" for future weather [3].

In the case of frequent extreme rainfall and untimely drainage, the infiltration of large amounts of rainwater causes changes in the pore structure and geometric characteristics inside tunnel soil. Tunnel projects are subjected to forces, deformation, and uneven settlement due to changes in each stratum's permeability characteristics and stress state, which adversely affect the function and safety of the tunnel [4–8]. Moreover, engineering practice also shows that rainfall and other natural disasters directly related to water are important factors leading to structural instability in subway tunnels, in turn inducing tunnel engineering accidents [4]. For example, in 2008, a major tunnel collapse of the Hangzhou Metro Line 1 occurred. The main reason for this accident was the continuous heavy rainfall during construction, which made the sandy soil more liquid. In 2008, the ground collapsed in Guangzhou Metro Line 5, and the Guangzhou Metro Line 2 Pazhou Tower section of the subway tunnel pavement collapsed in 2006. The person in charge of the rescue construction party at the scene said that days of rain had led to loose soil below the tunnel. Therefore, how to scientifically understand the occurrence and development of safety accidents in subway tunnel construction under extreme rainfall conditions and how to timely assess and predict the safety status of subway tunnels have become key issues in current research.

Presently, scholars' research on the influence of rainfall infiltration on tunnel safety has mainly focused on numerical simulations and model tests. Wang et al. [4] analyzed the effects of tunnel construction methods, geological conditions, and continuous rainfall on the ground surface and tunnel vault settlement. Cheng et al. [9], based on the dynamic finite element static-strength reduction method, analyzed the stability of loess tunnels under different rainfall conditions. Wang et al. [10] investigated the tunnel envelope's mechanical, hydraulic, and evolutionary characteristics under various precipitation scenarios. Zeng et al. [5] examined the changes in surface runoff, groundwater level, and water environment in the tunnel region using a hydrological monitoring system. Shi et al. [11] investigated how groundwater levels and precipitation affected leakage in tunnels with continuous arches. Lei et al.'s [6] analysis of field test data revealed the impact of precipitation on tunnel deformation. The impact of precipitation on earth pressure in shield tunnels built on expanding ground was evaluated by Chao et al. [12]. In less rich water, Xue et al. [7] investigated the effects of heavy precipitation on the stability and safety of tunnels. The influence of rainfall infiltration on the stability of existing offset twin tunnels with small diameters was researched by Chen et al. [13]. The above studies mainly focused on a specific load-bearing structure, such as tunnel vault, surrounding rock, and lining, and analyzed the influence of different rainfall intensities on the safety of a particular load-bearing structure of the tunnel. Few systematic safety evaluation studies have been conducted on the overall subway tunnel properties considering extreme rainfall. Grzegorz Wrzesiński et al. [14,15] applied artificial neural networks to evaluate the variation in undrained shear strength in cohesive soils due to rotation of principal stresses. Yum et al. [16] developed a new tunnel-centered natural disaster risk assessment method by performing multiple linear regression analyses on financial loss data generated from tunnel construction in Korea. Wang et al. [17] used the analytic hierarchy process to determine the baseline weights of tunnel construction dynamic risk assessment indexes. Liu et al. [18] proposed a support vector machine model based on a particle swarm algorithm for forecasting the safety risks in subway construction.

Tunnel safety evaluation involves numerous influencing factors, and numerous tunnel safety evaluation methods have emerged to effectively solve the problem of the complexity and uncertainty of factors.

Ou et al. [19] proposed a tunnel collapse risk assessment method based on case analysis and advanced geological prediction. Khosravizade et al. [20] employed the technique for order preference by similarity to an ideal solution method and the analytic hierarchy process decision making to evaluate the current risk in subway projects. An approach for evaluating the safety of tunnels was presented by Zhang et al. [21] and was based on case-based reasoning, geological forward prediction, and rough set theory. A tunnel safety evaluation model based on human reliability analysis was created by Kirytopoulos et al. [22].

Gkoumas et al. [23] employed the European PIARC-OECD quantitative risk assessment model for their risk study of road tunnels. A safety evaluation model for road tunnels was proposed by Kazaras et al. [24]. Road tunnel operations were assessed for safety, and the levels of tunnel danger were categorized by Schlosser et al. [25].

Bai et al. [26] established a soft risk evaluation model of rock tunnel deformation based on the standard cloud model. Wu et al. [27] proposed a risk evaluation model for sewage conveyance tunnels' construction phase based on a cloud model. Feng et al. [28] constructed a safety evaluation model for tunnel palm faces based on a hybrid particle swarm optimization neural network. Wen et al. [29] established a risk analysis model for tunnel burst water based on a fuzzy Bayesian network. Nezarat et al. [30] and Wang et al. [31] proposed a new model for quantitatively evaluating karst tunnel sudden water risk based on fuzzy analysis. Zhang et al. [32] used the mathematical attribute theory to establish a risk evaluation model for the slope stability of tunnel openings. He et al. [33] proposed a Bayesian network-based risk assessment method for large tunnel deformation.

Other scholars also conducted further research on tunnel safety evaluation. Zhou et al. [34] established a tunnel risk assessment system with several safety indices using the entire evaluation technique. A safety assessment index system and an extensive evaluation model for urban tunnels were devised by Zhou et al. [35]. A brand new approach for thoroughly assessing the danger of underwater shield tunnel construction was established by Wu et al. [36]. Lu et al. [37] developed a probabilistic index model, considering the correlation between signs of uncertainty and severe incidents while building subways. A quantitative measure was suggested by Malm-torp et al. [38] for evaluating the safety of road tunnels. Li et al.'s [39] drill-and-blast evaluation of the safety of large-span triplex tunnels was based on timely information from an integrated monitoring system.

From the above research results, it is clear that most studies evaluating tunnel safety have focused on determining how the tunnel is built and how it changes over time. Even though most of the proposed evaluation methods are helpful and are widely used in many fields, they still have some issues. (1) They fail to effectively solve the problems of the fuzziness and uncertainty of the evaluation indexes. (2) From an objective point of view, the degree of similarity between an evaluation index and a set of degrees of similarity is not fixed. Instead, it is within a specific range. The vital information about the range of change in the research object is left out of the current evaluation methods. (3) There are relatively few quantitative studies, many of which do not give specific safety evaluation levels.

To overcome the deficiencies mentioned above, we developed a safety evaluation model based on the multivariate linkage number and set pair analysis theory, considering extreme precipitation conditions. The model uses the benefits of set pair analysis to look at the system's deterministic and uncertain problems. Not only does it take into account how vague and uncertain the indicators are, but it also fixes problems with the traditional evaluation model, in which the values of the indicators are seen as fixed. The linear weighting idea is also introduced to perform the optimal combination of the indicator weights calculated by the improved hierarchical analysis and entropy weight method, which makes the weighting calculation more scientific and the evaluation results more accurate. The rest of this paper is organized as follows. Section 2 introduces the four methods used in this paper. Section 3 establishes a safety evaluation index system under extreme weather conditions and quantitative standards for index classification. In Section 4, the model is applied to evaluate the left line of a subway tunnel in an interval of an urban rail line (Line 2). Based on the results of Section 4, in Section 5, we further discuss the evaluation method proposed in this paper and present the overall conclusions.

## 2. Methods

### 2.1. The SPA Theory

Set pair analysis (SPA) is a mathematical theory first proposed by Zhao [40] in 1989. It is used to deal with the interaction between the certainty and uncertainty of a system. The SPA theory focuses on studying the interaction between certainty and uncertainty of a

given system under three aspects—common characteristics, opposite characteristics, and neither common nor opposite characteristics—to conduct a quantitative evaluation of the research object [31,41–44].

SPA has been widely used in many fields, such as tunnel collapse, tailings reservoir, and port ecological risk assessment. Using SPA, Chen et al. [43] provided a thorough evaluation technique of mountain tunnel collapse risk. A SPA quantitative risk assessment technique built on a fuzzy evaluation method was proposed by Shi et al. [44]. Li et al. [45] constructed a port ecological assessment model utilizing the SPA theory.

It is assumed that set pair $M$ is composed of the actual value of the evaluation index and the evaluation grade standard. There are $K$ indexes in set pair $M$, where $S$ is the standard part of the set pair, $F$ is the part that is neither expected nor opposite, and $P$ is the part that is opposite. They influence and restrain each other and transform into each other under certain conditions. The connection degree of a set pair $M$ can be expressed as:

$$\mu = \frac{S}{K} + \frac{F}{K}i + \frac{P}{K}j = a + bi + cj \tag{1}$$

where $a, b, c$ denote the identity, difference, and opposition of set pair $M$, respectively, with $a + b + c = 1$; $i$ and $j$ denote the coefficients of difference and opposition, respectively, with $i \in \begin{bmatrix} -1 & 1 \end{bmatrix}, j = -1$.

Sometimes, the SPA theory is necessary to deal with the connection degree in a diversified way. The expression is:

$$\mu = a + b_1 i_1 + b_2 i_2 + b_3 i_3 + \cdots + b_{k-2} i_{k-2} + cj \tag{2}$$

where $b_1, b_2, b_3, \cdots, b_{k-2}$ are the degrees of difference and $i_1, i_2, i_3, \cdots i_{k-2}$ are the coefficients of the degrees of difference, with $a + b_1 + b_2 + b_3 + \cdots + b_{k-2} + c = 1$.

The evaluation index may be classified into two categories based on the differences in characterization characteristics: the higher the evaluation index, the better, and the lower the evaluation index, the better. The connection degree calculation formulae are expressed as follows.

The more significant the assessment index, the better:

$$\mu = \begin{cases} 1 + 0i_1 + 0i_2 + 0i_3 + 0j, x_c \geq s_1 \\ \frac{x_c - s_2}{s_1 - s_2} + \frac{s_1 - x_c}{s_1 - s_2}i_1 + 0i_2 + 0i_3 + 0j, s_2 \leq x_c \leq s_1 \\ 0 + \frac{x_c - s_3}{s_2 - s_3}i_1 + \frac{s_2 - x_c}{s_2 - s_3}i_2 + 0i_3 + 0j, s_3 \leq x_c \leq s_2 \\ 0 + 0i_1 + \frac{x_c - s_4}{s_3 - s_4}i_2 + \frac{s_3 - x_c}{s_3 - s_4}i_3 + 0j, s_4 \leq x_c \leq s_3 \\ 0 + 0i_1 + 0i_2 + \frac{x_c - s_5}{s_4 - s_5}i_3 + \frac{s_4 - x_c}{s_4 - s_5}j, s_5 \leq x_c \leq s_4 \\ 0 + 0i_1 + 0i_2 + 0i_3 + 1j, x_c < s_5 \end{cases} \tag{3}$$

The better the assessment index, the smaller it is:

$$\mu = \begin{cases} 1 + 0i_1 + 0i_2 + 0i_3 + 0j, x_c < s_1 \\ \frac{s_2 - x_c}{s_2 - s_1} + \frac{x_c - s_1}{s_2 - s_1}i_1 + 0i_2 + 0i_3 + 0j, s_1 < x_c \leq s_2 \\ 0 + \frac{s_3 - x_c}{s_3 - s_2}i_1 + \frac{x_c - s_2}{s_3 - s_2}i_2 + 0i_3 + 0j, s_2 < x_c \leq s_3 \\ 0 + 0i_1 + \frac{s_4 - x_c}{s_4 - s_3}i_2 + \frac{x_c - s_3}{s_4 - s_3}i_3 + 0j, s_3 < x_c \leq s_4 \\ 0 + 0i_1 + 0i_2 + \frac{s_5 - x_c}{s_5 - s_4}i_3 + \frac{x_c - s_4}{s_5 - s_4}j, s_4 < x_c \leq s_5 \\ 0 + 0i_1 + 0i_2 + 0i_3 + 1j, x_c > s_5 \end{cases} \tag{4}$$

where $x_c$ is the evaluation index value, and $s_1, s_2, s_3, s_4, s_5$ are the evaluation index grade critical values.

### 2.2. Index Weights

Weights are divided into subjective weights and objective weights. In this study, the improved analytic hierarchy process (IAHP) and entropy weight method (EWM) are used

to determine the subjective and objective weights, respectively. In addition, the linear weighting method is used to obtain the total weights of each factor. Furthermore, the consistency test is carried out using the distance function to improve the reliability of the evaluation results.

### 2.2.1. The IAHP Approach

The analytic hierarchy process (AHP) is a weight decision analysis method proposed by American operations research scientist Saaty in the early 1970s [46]. Its application process can be briefly divided into two steps. The first step is constructing the evaluation system according to the research problems. The second step is to select an accurate and reasonable calculation method and carry out a weight analysis combined with the constructed evaluation system.

To avoid the defects of the analytic hierarchy process, such as a large amount of calculation when there are many evaluation indexes and the inability to ensure the accuracy of the analysis results effectively, many scholars have improved it to varying degrees. Liang et al. [47] simplified the tedious calculation process of the traditional analytic hierarchy process by constructing the optimal transfer matrix. Zuo [48] improved the nine-scale theory in the traditional analytic hierarchy process by designing the three-scale theory, which solved the problems of complex data and the inconsistency of the analysis results. We combined the above two improvement methods and then evaluated the safety of subway tunnels. The specific steps are as follows.

Step 1: Construct the comparison matrix by applying the three-scale theory, which is marked as $A$:

$$A = \begin{bmatrix} a_{11} & a_{12} & \cdots & a_{1n} \\ a_{21} & a_{22} & \cdots & a_{2n} \\ \vdots & \vdots & \ddots & \vdots \\ a_{n1} & a_{n2} & \cdots & a_{nn} \end{bmatrix} \tag{5}$$

When the three-scale theory is adopted, the specific value rules of each element $a_{ij}$ in the comparison matrix $A$ are shown in Table 1.

**Table 1.** Value-taking rules of each element of the comparison matrix $A$.

| Value of $a_{ij}$ | Definition |
| --- | --- |
| 0 | Indicates that $j$ is more critical than $i$ |
| 1 | Indicates that $i$ is as crucial as $j$ |
| 2 | Indicates that $i$ is more critical than $j$ |

Step 2: Construct a judgment matrix, which is marked as $B$. The element $b_{ij}$ of the matrix $B$ is calculated as follows:

$$bij = \begin{cases} \frac{r_i - r_j}{r_{max} - r_{min}}(k_m - 1) + 1, r_i \geq r_j \\ \frac{1}{\frac{|r_i - r_j|}{r_{max} - r_{min}}(k_m-1)+1}, r_i < r_j \end{cases}, k_m = \frac{r_{max}}{r_{min}} \tag{6}$$

Step 3: Construct an optimal transfer matrix, which is denoted as $C$. The element $c_{ij}$ of the matrix $C$ is calculated as follows:

$$\begin{cases} c^* = lgb_{ij} \\ c_{ij} = \frac{\sum_{k=1}^{n}\left(c_{ik}^* - c_{kj}^*\right)}{n} \end{cases} \tag{7}$$

Step 4: Construct a quasi-optimal consistent matrix of judgment matrix $B$, which is denoted as $D$. The element $d_{ij}$ of the matrix $D$ is calculated as follows:

$$d_{ij} = 10^{c_{ij}} \tag{8}$$

Step 5: Normalize each column of the quasi-optimal consistent matrix $D$ of judgment matrix $B$:

$$d_{ij}^* = \frac{d_{ij}}{\sum_{k=1}^{n} d_{kj}} \qquad (9)$$

Step 6: Add the normalized quasi-optimal consistent matrix according to the direction of the row; then, divide each element in the vector obtained by summation by $n$ to obtain the eigenvector of the quasi-optimal consistent matrix $D$, that is, the subjective weight vector:

$$w_1 = \frac{\sum_{k=1}^{n} d_{ik}^*}{n} \qquad (10)$$

### 2.2.2. EWM Approach

Entropy was introduced into information theory by Shannon in 1948 to describe the disorder degree of information. EWM is a method to calculate the objective weight; the more significant the index's entropy, the smaller the amount of information and weight and vice versa [49,50]. The specific steps are as follows.

Step 1: An initial decision matrix is established, denoted as $X$:

$$X = \begin{bmatrix} x_{11} & x_{12} & \dots & x_{1m} \\ x_{21} & x_{22} & \dots & x_{2m} \\ \vdots & \vdots & \ddots & \vdots \\ x_{n1} & x_{n2} & \dots & x_{nm} \end{bmatrix} \qquad (11)$$

where $n$ represents the total number of evaluation objects to be evaluated; $m$ represents the total number of evaluation indexes and the $j$th index weight of the $i$th evaluation object.

Step 2: Normalization processing

When the entropy weight method is used to calculate the objective weight, the dimensional units of each evaluation index need to be normalized to compare the evaluation indexes with each other.

Income-type index:

$$y_{ij} = \frac{x_{ij} - min_j(x_{ij})}{max_j(x_{ij}) - min_j(x_{ij})} \qquad (12)$$

Cost-type index:

$$y_{ij} = \frac{max_j(x_{ij}) - x_{ij}}{max_j(x_{ij}) - min_j(x_{ij})} \qquad (13)$$

Step 3: Information entropy calculation

$$\begin{cases} p_{ij} = \frac{1+y_{ij}}{\sum_{i=1}^{n}(1+y_{ij})} \\ E_j = -\frac{1}{ln} \sum_{i=1}^{n} \left( p_{ij} ln p_{ij} \right) \end{cases} \qquad (14)$$

Step 4: Calculation of the objective weight

$$W_2 = \frac{1 - E_j}{n - \sum_{j=i}^{m} E_j} \qquad (15)$$

### 2.2.3. Comprehensive Weight

The subjective weighting method represents the intuitive knowledge of experts, and there are many human interference factors. The objective weighting method represents the objective law of the measured data. Therefore, combining the two weight methods is vital to obtain a more scientific and reasonable comprehensive weight. We combined the subjective and objective weights with a weighted linear combination.

Firstly, the distance function $d(w_1, w_2)$ is used to check the consistency of the determined index weights to avoid contradiction between the weights determined by different weight calculation methods. If $d(w_1, w_2) \in [-1, 1]$, the objective and subjective weights are combined based on linear weighting to obtain the optimal comprehensive weight. Otherwise, the weight of each indicator needs to be recalculated [51]:

$$d(w_1, w_2) = \sqrt{\sum_{K=1}^{N}(w_1 - w_2)^2} \tag{16}$$

$$\begin{cases} d^2(w_1, w_2) = (p - q)^2 \\ p + q = 1 \end{cases} \tag{17}$$

$$w = qw1 + pw_2 \tag{18}$$

where $q$ is the subjective weight coefficient calculated using the improved hierarchical method, and $p$ is the objective weight coefficient calculated using the entropy weight method.

### 2.3. Comprehensive Correlation Degree Model

Assuming that the eigenvector matrix $E = (1, i_1, i_2, i_3, j)^T$, that the weight set of the indexes is $W = (w_1, w_2, \cdots, w_n)$, and that the matrix formed by the single-index connection degree is R, then the comprehensive connection degree of set pair M can be expressed as Equation (19).

$$\mu = WRE \tag{19}$$

### 2.4. Set Pair Potential Analysis

The set pair potential represents the degree of connection between the contract differences and inversions of the two sets, which can reflect the development trend of the set pairs under certain conditions, as shown in Table 2.

**Table 2.** Rank criteria of set pair potential.

| Situation | Gradation | Discriminant Condition | Coordination | Development Trend |
|---|---|---|---|---|
| Same potential | Quasi same potential | a > 0, b = 0 | Stronger identity | Stronger improvement trend |
| | Strong same potential | a > c, c > b | Strong identity | Strong improvement trend |
| | Weak same potential | a > c, a > c > b | Weaker identity | Weaker improvement trend |
| | Micro same potential | a = c, b > a | Weak identity | Weak improvement trend |
| Balance of power | Quasi balance of power | a = 0, b = 0 | Strong stability | Weak improvement trend |
| | Strong balance of power | a = c, a > b >0 | Stronger stability | Weaker improvement trend |
| | Weak balance of power | a = c, b = a | Weaker stability | Stronger improvement trend |
| | Micro balance of power | a = c, b > a | Weak stability | Strong improvement trend |
| Counter potential | Quasi counter potential | a < 0, b = 0 | Stronger opposition | Stronger improvement trend |
| | Strong counter potential | a < c, 0 < b < a | Strong opposition | Strong improvement trend |
| | Weak counter potential | a < c, b > a, b < c | Weaker opposition | Weaker improvement trend |
| | Micro counter potential | a < c, b > c | Weak opposition | Weak improvement trend |

### 2.5. Procedure of Safety Evaluation Based on SPA and IAHP–EWM

According to the set pair analysis method and the combination weighting method introduced in Sections 2.1–2.4, the proposed safety assessment process of subway tunnels under extreme weather conditions is shown in Figure 1.

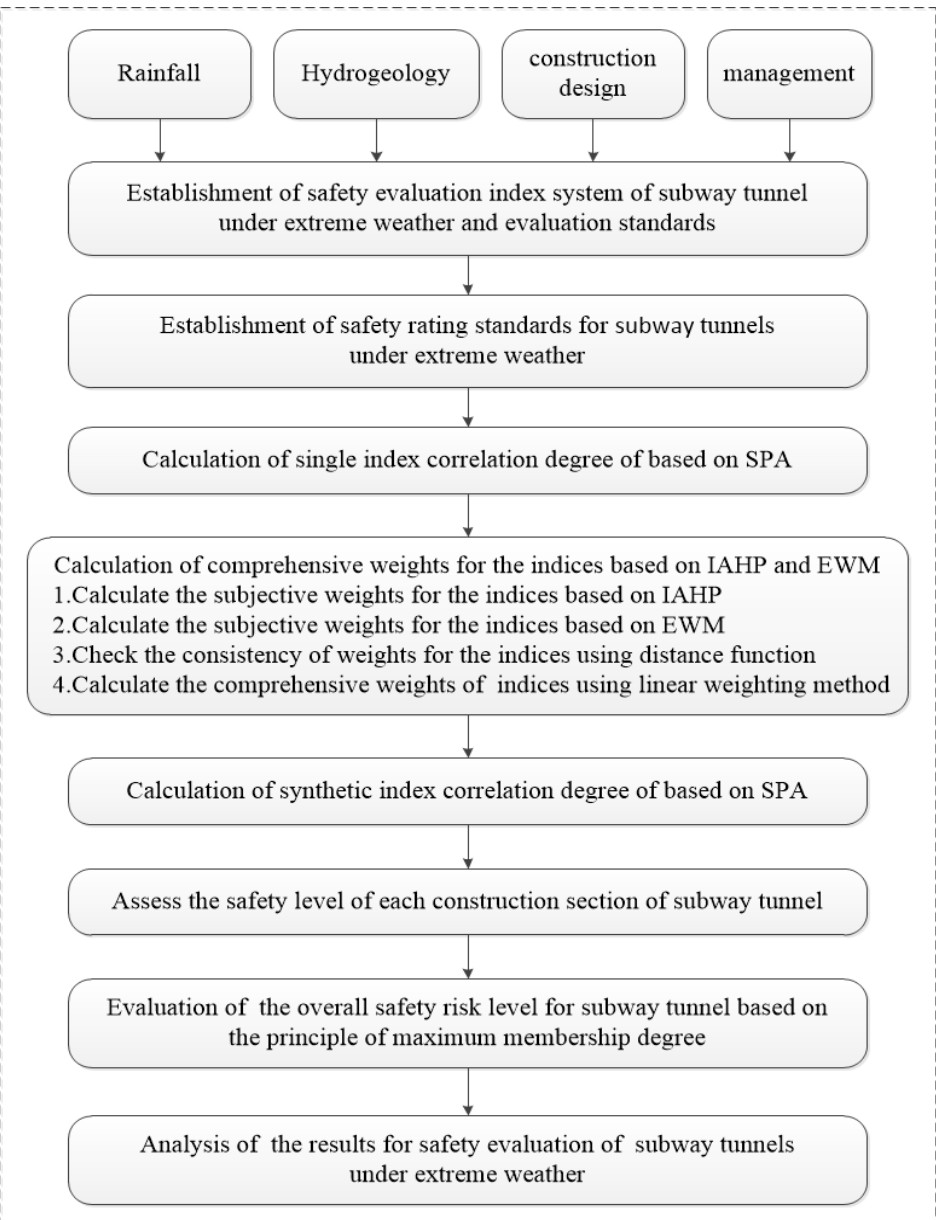

**Figure 1.** Flow chart of risk assessment of subway tunnel safety under extreme weather conditions.

## 3. Safety Evaluation Model

The safety problem of subway tunnel construction under extreme rainfall is caused by the combination of external environmental changes caused and unsafe internal construction conditions. Under extreme rainfall conditions, rainwater first comes into contact with the surface soil, and the shallow surface first reaches saturation. With rainwater infiltration, rainwater infiltrates via surface crevices and converges at the bottom of the crevices to form a saturated zone, and the pore pressure value within the saturated zone is higher than that of the soil at other locations. Therefore, the soil at the top of the tunnel settles, while the soil at the bottom is disturbed very little by the rainfall conditions, causing shear damage to the tunnel envelope soil. The increase in rainfall duration expands the range of rainwater-wetted soil.

Moreover, in the case of poor drainage, long-term rainfall makes the vertical tunnel displacement more significant than the control value, leading to a particular safety hazard. The increase in rainfall intensity accelerates the rate of soil moisture content. When the rainfall intensity exceeds a specific value, rainwater is not thoroughly infiltrated, with the

ground surface above the water table becoming the primary accumulation location after rainwater infiltration. In these soil locations, pore water pressure increases, and matrix suction decreases, weakening the shear strength of the tunnel soil. A short period of modest rainfall does not significantly affect the groundwater level of the soil, but with the increase in rainfall load, the groundwater level rises, and the bearing capacity of the soil decreases while the surface settlement increases. At the same time, the swelling force of the soil increases due to the increase in water content, and the range of plastic zone where shear damage occurs in the tunnel soil gradually increases [4,10,16]. The internal construction status, influenced by factors such as the proficiency of professional skills of the construction personnel, the strength of monitoring and detection, and the perfection of the construction scheme, also affects the safety and stability of the subway tunnel structure.

Based on the above, we combined saturated–unsaturated seepage theory and studied the detrimental impacts of changes in physical parameters, such as water content and soil permeability, on the structural system of the subway tunnel under extreme rainy conditions for a variety of rainfall volumes, intensities, and durations, laying the groundwork for building a safety evaluation model of a subway tunnel in the event of heavy rain. The technical route of this work is detailed below.

Step 1: Firstly, the construction of subway tunnels in China and the development trend of extreme weather are introduced. Furthermore, it is pointed out that subway tunnels are more affected by extreme rainfall weather, which leads to the frequent occurrence of subway tunnel construction accidents, making the research topic of the safety impact of extreme rainfall weather on subway tunnels an important issue in recent years. Secondly, the current status of research on tunnel safety evaluation is analyzed by reviewing the literature.

Step 2: Based on the research achievement and numerical simulations conducted by domestic and foreign scholars on the influence of rainfall infiltration on the safety of subway tunnels, the risk factors affecting the safety of subway tunnels during extreme rainfall are identified and quantified, and a safety evaluation index system and index safety classification criteria are established.

Step 3: We apply the improved hierarchical analysis method and entropy weight method to calculate the index weights and check the consistency of the index weights using the distance function. Meanwhile, we introduce the idea of linear weighting to combine the subjective and objective weights. Furthermore, we construct the safety evaluation model of a subway tunnel under extreme rainfall weather conditions based on the multivariate linkage number and set pair analysis theory.

Step 4: A specific interval of the tunnel of a city rail line (Line 2) is taken as a research case to verify the scientificity and validity of the evaluation model.

### 3.1. Safety Evaluation Index System

This paper is based on the description of the main risk events that may occur during the construction of urban rail transit projects according to "Risk Assessment Guide for Urban Rail Transit Projects", "Code of Practice for the Construction Management of Urban Rail Transit Underground Projects" GB50652-2011, and other related codes. The factors affecting the safety of subway tunnels under extreme rainfall weather conditions are divided into accident-inducing rainfall factors, accident-nurturing hydrogeological factors and management factors, and construction design factors to prevent accidents.

To comprehensively identify the risk factors of subway tunnel construction under extreme rainfall conditions, the above four influencing factors are used as the first-level evaluation indexes, and further detailed classification is conducted. For the influence of the "external environment", the key factors affecting the safety of subway tunnels under extreme rainfall conditions are selected according to the literature review and case study. For the influence of "internal construction status", the evaluation indexes are obtained by interviewing subway practitioners and consulting underground engineering experts. Finally, the Delphi method is used to process the obtained evaluation indexes and construct a safety evaluation index system, as shown in Table 3. Moreover, based on the saturated–

unsaturated seepage theory, the relationship between the indicators and the overall effect on the safety state of the subway tunnel is analyzed under the three aspects of rainfall volume, rainfall intensity, and rainfall duration, and the cause–effect relationship diagram is shown in Figure 2.

**Table 3.** Safety evaluation index system.

| Primary Indicators | Secondary Indicators | Number | Source |
|---|---|---|---|
| Rainfall $B_1$ | Amount of Rainfall (mm/d) Surface water depth (mm) | $C_{11}$ $C_{12}$ | [9,10] [5] |
| Hydrogeology $B_2$ | Groundwater level (m) Poisson's ratio Cohesion (KPa) Angle of internal friction (°) Water content (%) Permeability coefficient (cm/s) | $C_{21}$ $C_{22}$ $C_{23}$ $C_{24}$ $C_{25}$ $C_{26}$ | [11,52] [9,10] [9,10] [10,13] [4,12] [5] |
| Construction design $B_3$ | Tunnel depth (m) Tunnel diameter (m) Lining thickness (mm) | $C_{31}$ $C_{32}$ $C_{33}$ | [6,52] [4] [6,11] |
| Management $B_4$ | Monitoring and testing efforts Construction organization and design Safety organization and system Professional skills of construction personnel | $C_{41}$ $C_{42}$ $C_{43}$ $C_{44}$ | —— —— —— —— |

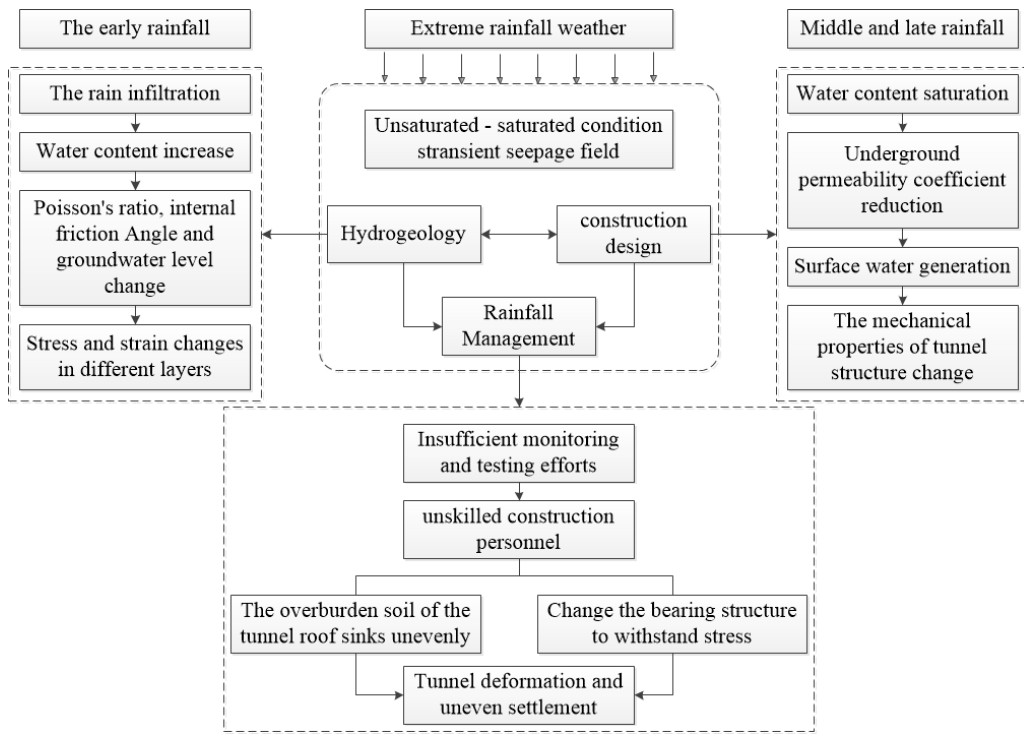

**Figure 2.** Causal diagram of influencing factors.

(1) Rainfall ($B_1$) and Hydrogeology ($B_2$): In the process of extreme rainfall, at the beginning of rainfall, due to the low rainfall intensity and rainfall, rainwater can infiltrate completely, and the water content ($C_{25}$) of the soil increases gradually, while hydrogeological conditions such as Poisson's ratio ($C_{22}$), cohesion ($C_{23}$), and internal friction angle ($C_{24}$) also change accordingly. With the passage of rainfall time, the groundwater level ($C_{21}$) rises gradually, which makes the subway tunnel a complex seepage field between

saturated and unsaturated conditions, resulting in corresponding changes in the stress and strain of each stratum and the mechanical characteristics of the tunnel structure, so that the safety and stability of the subway tunnel structure are in a complex state that changes with time and space.

In the middle and later periods of extreme rainfall, the water content of the covering soil on top of the tunnel gradually reaches saturation, and the permeability coefficient ($C_{26}$) of the soil gradually decreases until the infiltration rate of the soil is less than the rainfall intensity; then, the surface begins to accumulate water with the increase in rainfall time and rainfall ($C_{11}$). The surface ponding depth ($C_{12}$) increases accordingly. Under the actions of surface water's softening and erosion, soil's bearing capacity decreases gradually, further deteriorating the subway tunnel structure's deformation and mechanical characteristics.

(2) Engineering design ($B_3$): In the tunnel structure design parameters, the tunnel burial depth ($C_{31}$), tunnel diameter ($C_{32}$), and lining thickness ($C_{33}$) are the main parameters to ensure the safety and stability of the tunnel structure, which are determined by theoretical calculation according to the geological conditions and load pressure of the tunnel. However, in the dynamic process of extreme rainfall and saturated–unsaturated rainwater infiltration, the hydrogeological conditions and surface environment change with the gradual increase in the soil moisture content, which leads to the change in the mechanical properties of each soil layer and soil subsidence. The stress of the bearing structure, such as the lining and surrounding rock of the subway tunnel, presents varying degrees of change trend under the action of differential settlement and extrusion of the covering soil on top of the tunnel, which results in deformation and differential settlement of the tunnel.

(3) Management factors ($B_4$): Based on the quality management system 4M1E standard, it is believed that incomprehensive and non-specific risk monitoring methods ($C_{41}$), construction personnel irregularities ($C_{42}$), and poor construction plans ($C_{43}$) in underpass-tunnel construction are subjective factors in the generation of safety accidents.

### 3.2. Grade Division Standard of Evaluation Indexes

From an objective point of view, the degree of affiliation of each evaluation indicator to the specified set varies within a specific range. Therefore, in this section, we present the results of the analysis of the range of safety level variations in the safety evaluation indicators for subway tunnels under extreme rainfall weather conditions proposed in Section 2.1.

According to "Risk Occurrence Possibility and Risk Loss Grade Standard in the Code for Risk Management of Underground Engineering Construction of Urban Rail Transit", we divided the safety grade of subway tunnel construction under extreme rainfall weather conditions into five levels. The proposed safety evaluation index system for subway tunnels under extreme rainfall involves objective evaluation factors (e.g., construction and geological) and subjective evaluation factors (e.g., management). Objective factors were measured by the previous experimental and simulation studies on the evolution of tunnel safety and stability considering a particular factor. The quantitative grading scale of objective factors is determined based on the statistical idea of researching many literature studies. Relevant experts determine subjective factors. Table 4 shows the scoring details of subjective factors based on a 100-mark system (0–100). The intervals for subjective factors are recognized by combining with engineering practices and theoretical analysis. The grading criteria of the evaluation indexes are established, as shown in Table 4.

### 3.3. Grades of Safety Assessment

The safety level of the assessment indices is separated into five levels, as indicated in Table 4. The safety level of the subway tunnel is also separated into five categories for the final evaluation findings to conform to them. Since the range $[-1, 1]$ is chosen for the set of comprehensive connections, we built the security level determination table, shown in Table 5, based on the "principle of equal division" for the range $[-1, 1]$.

**Table 4.** Classification criteria of evaluation indexes.

| Index | Grading Standard | | | | |
|---|---|---|---|---|---|
| | Extremely Safe | Safe | Basically Safe | Unsafe | Extremely Unsafe |
| $C_{11}$ | < 25 | 25–50 | 50–100 | 100–250 | > 250 |
| $C_{12}$ | 0–400 | 400–800 | 800–1200 | 1200–1600 | 1600–2000 |
| $C_{21}$ | > 6 | 5–6 | 3–5 | 2–3 | < 2 |
| $C_{22}$ | 0–0.2 | 0.2–0.25 | 0.25–0.3 | 0.3–0.35 | 0.35–0.5 |
| $C_{23}$ | > 5.5 | 2.1–5.5 | 0.7–2.1 | 0.2–0.7 | 0–0.2 |
| $C_{24}$ | 60–90 | 50–60 | 39–50 | 27–39 | 0–27 |
| $C_{25}$ | 0–5 | 5–10 | 10–15 | 15–20 | 20–30 |
| $C_{26}$ | $< 10^{-6}$ | $10^{-6}$–$10^{-4}$ | $10^{-4}$–$10^{-3}$ | $10^{-3}$–$10^{-1}$ | $> 10^{-1}$ |
| $C_{31}$ | 32–40 | 24–32 | 16–24 | 8–16 | > 8 |
| $C_{32}$ | 0–6 | 6–9 | 9–12 | 12–15 | > 15 |
| $C_{33}$ | 700–800 | 600–700 | 400–600 | 200–400 | 50–200 |
| $C_{41}$ | Very high (80–100) | High (60–80) | Normal (40–60) | Low (20–40) | Very low (0–20) |
| $C_{42}$ | Perfect (80–100) | Good (60–80) | Normal (40–60) | Poor (20–40) | Worst (0–20) |
| $C_{43}$ | Perfect (80–100) | Good (60–80) | Normal (40–60) | Poor (20–40) | Worst (0–20) |
| $C_{44}$ | Very high (80–100) | High (60–80) | Normal (40–60) | Low (20–40) | Very low (0–20) |

**Table 5.** Determination of safety and risk levels.

| Judgment Interval | (0.6,1] | (0.2,0.6] | (−0.2,0.2] | (−0.6,−0.2] | [−1,−0.6] |
|---|---|---|---|---|---|
| Safety Level | I | II | III | IV | V |
| Risk Level | very low | low | medium | high | very high |

Level I means that the safety risk is low and can be ignored, and the tunnel construction can be carried out according to the established tunnel construction plan. Level II means that the safety risk is low and within the expected range and that the tunnel construction can be carried out according to the established plan, but the frequency of monitoring and measurement must be increased. Level III means that the safety risk is medium and within the controllable range, and preventive measures can be taken for potential risk sources to reduce the probability of accidents. Level IV represents a high risk beyond the expected range and requires safety measures and increased daily monitoring. Level V represents a very high risk, which may lead to a wide range of structural instabilities and requires real-time monitoring and testing of the tunnel and reasonable measures to maintain the stability of the tunnel.

## 4. Case Study

On the south bank of the Yellow River, in the valley basin, is a portion of the Line 2 subway tube. The geomorphological unit is a piece of the first terrace of the Yellow River. The line is parallel to the earth. Quaternary Holocene artificial fill, alluvial loess-like silt, fine silty sand, pebbles, and Lower Tertiary sandstone make up the site layer. The length of the left line is 661.143 m. Considering the longitudinal section, the left line of the section goes downhill at gradients of 28‰ and 8.0‰ and then goes uphill at a gradient of 21.824‰ to the next station, mainly passing through the pebble layer and strongly weathered sandstone layer.

According to the geological exploration report, the water-bearing layers in this area are the pebble layer, strongly weathered sandstone layer, and weathered sandstone layer, among which the pebble layer has high density, a high percolation rate, and high water content, and is the main water-bearing layer in the construction site. Therefore, we divided the interval tunnel into three construction sections, as shown in Figure 3, denoted as $L_1$, $L_2$,

and $L_3$, based on the division of water-bearing layers in the foundation soil layer; these are taken as the evaluation objects for the safety evaluation of this interval tunnel.

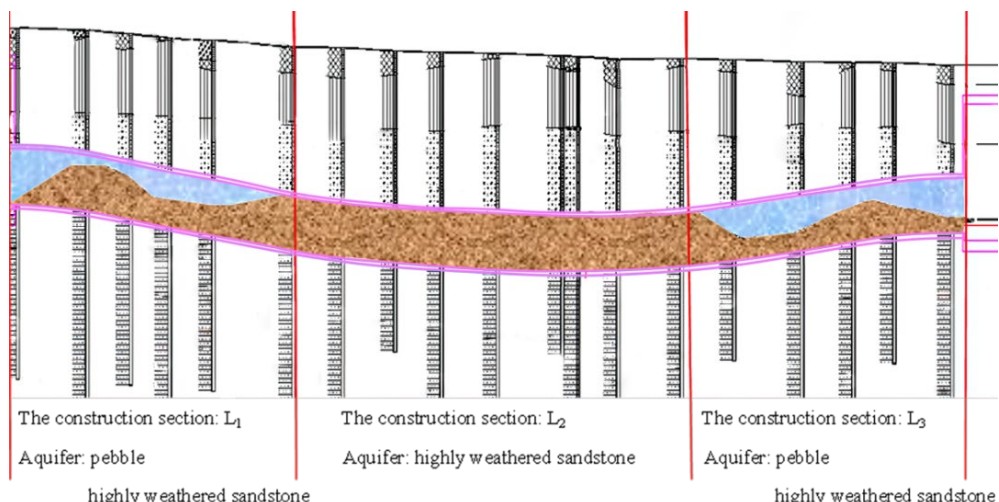

**Figure 3.** Geological longitudinal section of interval.

*4.1. Data Acquisition*

Based on the evaluation index system established in Section 2.1, the geological survey report, and the subway tunnel engineering design data, the index parameters required for the collected $L_1$, $L_2$, and $L_3$ are shown in Table 6.

**Table 6.** Evaluation index parameters.

| Index | Object of Evaluation | | |
|---|---|---|---|
| | $L_1$ | $L_2$ | $L_3$ |
| $C_{11}$ | 45 | 45 | 45 |
| $C_{12}$ | 13 | 20 | 16 |
| $C_{21}$ | 9.1 | 8.3 | 9.0 |
| $C_{22}$ | 0.25 | 0.28 | 0.23 |
| $C_{23}$ | 12 | 25 | 10 |
| $C_{24}$ | 37 | 35 | 38 |
| $C_{25}$ | 19.7 | 16.2 | 19.7 |
| $C_{26}$ | 0.0405 | 0.0035 | 0.0347 |
| $C_{31}$ | 15 | 15.5 | 15 |
| $C_{32}$ | 6.2 | 5.5 | 6.0 |
| $C_{33}$ | 460 | 450 | 460 |
| $C_{41}$ | High (75) | High (75) | High (75) |
| $C_{42}$ | Good (75.4) | Good (74.8) | Good (75.4) |
| $C_{43}$ | Good (75) | Good (75) | Good (75) |
| $C_{44}$ | High (75.6) | High (75.4) | High (75.4) |

*4.2. Safety Assessment*

According to the flow chart of the safety assessment of subway tunnel construction considering extreme weather shown in Figure 1 in Section 2.5, the safety level of the subway tunnel is assessed in this section, and the assessment process is described in the following subsections.

4.2.1. Correlation Calculation

The single-index connection degree of $L_1$, $L_2$, and $L_3$ are computed using Equations (1)–(4), and the calculation results are shown in Table 7.

**Table 7.** Single-index connection degree.

| Index | L$_1$ | | | | | L$_2$ | | | | | L$_3$ | | | | |
|---|---|---|---|---|---|---|---|---|---|---|---|---|---|---|---|
| | a | b$_1$ | b$_2$ | b$_3$ | c | a | b$_1$ | b$_2$ | b$_3$ | c | a | b$_1$ | b$_2$ | b$_3$ | c |
| C$_{11}$ | 0.000 | 0.200 | 0.800 | 0.000 | 0.000 | 0.000 | 0.200 | 0.800 | 0.000 | 0.000 | 0.000 | 0.200 | 0.800 | 0.000 | 0.000 |
| C$_{12}$ | 0.968 | 0.032 | 0.000 | 0.000 | 0.000 | 0.950 | 0.050 | 0.000 | 0.000 | 0.000 | 0.960 | 0.040 | 0.000 | 0.000 | 0.000 |
| C$_{21}$ | 1.000 | 0.000 | 0.000 | 0.000 | 0.000 | 1.000 | 0.000 | 0.000 | 0.000 | 0.000 | 1.000 | 0.000 | 0.000 | 0.000 | 0.000 |
| C$_{22}$ | 0.000 | 1.000 | 0.000 | 0.000 | 0.000 | 0.000 | 0.400 | 0.600 | 0.000 | 0.000 | 0.400 | 0.600 | 0.000 | 0.000 | 0.000 |
| C$_{23}$ | 1.000 | 0.000 | 0.000 | 0.000 | 0.000 | 1.000 | 0.000 | 0.000 | 0.000 | 0.000 | 1.000 | 0.000 | 0.000 | 0.000 | 0.000 |
| C$_{24}$ | 0.000 | 0.000 | 0.000 | 0.833 | 0.167 | 0.000 | 0.000 | 0.000 | 0.670 | 0.330 | 0.000 | 0.000 | 0.000 | 0.920 | 0.080 |
| C$_{25}$ | 0.000 | 0.000 | 0.000 | 0.060 | 0.940 | 0.000 | 0.000 | 0.000 | 0.760 | 0.240 | 0.000 | 0.000 | 0.000 | 0.060 | 0.940 |
| C$_{26}$ | 0.000 | 0.000 | 0.000 | 0.600 | 0.400 | 0.000 | 0.000 | 0.000 | 0.970 | 0.030 | 0.000 | 0.000 | 0.000 | 0.660 | 0.340 |
| C$_{31}$ | 0.000 | 0.000 | 0.000 | 0.875 | 0.125 | 0.000 | 0.000 | 0.000 | 0.940 | 0.060 | 0.000 | 0.000 | 0.000 | 0.880 | 0.120 |
| C$_{32}$ | 0.000 | 0.930 | 0.070 | 0.000 | 0.000 | 0.080 | 0.920 | 0.000 | 0.000 | 0.000 | 0.000 | 1.000 | 0.000 | 0.000 | 0.000 |
| C$_{33}$ | 0.000 | 0.000 | 0.300 | 0.700 | 0.000 | 0.000 | 0.000 | 0.250 | 0.750 | 0.000 | 0.000 | 0.000 | 0.300 | 0.700 | 0.000 |
| C$_{41}$ | 0.000 | 0.750 | 0.250 | 0.000 | 0.000 | 0.000 | 0.750 | 0.250 | 0.000 | 0.000 | 0.000 | 0.000 | 0.750 | 0.250 | 0.000 |
| C$_{42}$ | 0.000 | 0.770 | 0.230 | 0.000 | 0.000 | 0.000 | 0.740 | 0.260 | 0.000 | 0.000 | 0.000 | 0.000 | 0.770 | 0.230 | 0.000 |
| C$_{43}$ | 0.000 | 0.750 | 0.250 | 0.000 | 0.000 | 0.000 | 0.750 | 0.250 | 0.000 | 0.000 | 0.000 | 0.000 | 0.750 | 0.250 | 0.000 |
| C$_{44}$ | 0.000 | 0.780 | 0.220 | 0.000 | 0.000 | 0.000 | 0.770 | 0.230 | 0.000 | 0.000 | 0.000 | 0.000 | 0.770 | 0.230 | 0.000 |

4.2.2. Weight Calculation

1.     Subjective Weight Calculation

According to the established risk assessment model of subway tunnel safety under extreme weather conditions, experts compare the primary and secondary indicators according to the rules for taking the values of each element of the comparison matrix *A* in Table 1. The comparison matrix *A* is shown in the matrix (20).

$$
A = \begin{bmatrix}
2 & 2 & 2 & 2 & 1 & 2 & 1 & 1 & 2 & 2 & 2 & 2 & 2 & 2 & 2 \\
0 & 0 & 0 & 0 & 0 & 1 & 0 & 0 & 0 & 0 & 0 & 0 & 1 & 1 & 2 \\
1 & 2 & 2 & 2 & 1 & 2 & 2 & 0 & 2 & 2 & 2 & 1 & 2 & 2 & 1 \\
0 & 1 & 0 & 0 & 0 & 2 & 0 & 0 & 2 & 2 & 2 & 0 & 1 & 1 & 0 \\
0 & 2 & 1 & 2 & 0 & 2 & 0 & 0 & 2 & 2 & 2 & 0 & 1 & 1 & 0 \\
0 & 2 & 0 & 1 & 0 & 2 & 0 & 0 & 2 & 2 & 2 & 0 & 1 & 1 & 0 \\
1 & 2 & 2 & 2 & 1 & 2 & 0 & 1 & 2 & 2 & 2 & 1 & 2 & 2 & 1 \\
0 & 2 & 2 & 2 & 2 & 2 & 1 & 1 & 2 & 2 & 2 & 1 & 2 & 2 & 0 \\
0 & 0 & 0 & 0 & 0 & 2 & 0 & 0 & 1 & 0 & 0 & 1 & 2 & 2 & 1 \\
0 & 0 & 0 & 0 & 0 & 2 & 0 & 0 & 2 & 1 & 0 & 0 & 1 & 1 & 0 \\
0 & 0 & 0 & 0 & 0 & 2 & 0 & 0 & 2 & 2 & 1 & 1 & 2 & 2 & 1 \\
1 & 2 & 2 & 2 & 1 & 2 & 1 & 0 & 1 & 2 & 1 & 1 & 2 & 2 & 0 \\
0 & 1 & 1 & 1 & 0 & 1 & 0 & 0 & 0 & 1 & 0 & 0 & 1 & 1 & 0 \\
0 & 1 & 1 & 1 & 0 & 1 & 0 & 0 & 0 & 1 & 0 & 0 & 1 & 1 & 0 \\
1 & 2 & 2 & 2 & 1 & 0 & 2 & 0 & 1 & 2 & 1 & 2 & 2 & 2 & 1
\end{bmatrix} \tag{20}
$$

Secondly, Equation (6) was used to construct the judgment matrix *B*, as shown in the matrix (21).

$$
\begin{bmatrix}
4.900 & 21.800 & 16.600 & 19.200 & 6.200 & 29.600 & 6.200 & 1.000 & 24.400 & 27.000 & 19.200 & 10.100 & 27.000 & 27.000 & 8.800 \\
0.039 & 0.114 & 0.071 & 0.088 & 0.041 & 1.000 & 0.041 & 0.034 & 0.161 & 0.278 & 0.088 & 0.049 & 0.278 & 0.278 & 0.046 \\
1.000 & 17.900 & 12.700 & 15.300 & 2.300 & 25.700 & 2.300 & 0.204 & 20.500 & 23.100 & 15.300 & 6.200 & 23.100 & 23.100 & 4.900 \\
0.056 & 1.000 & 0.161 & 0.278 & 0.060 & 8.800 & 0.060 & 0.046 & 3.600 & 6.200 & 0.278 & 0.079 & 6.200 & 6.200 & 0.071 \\
0.079 & 6.200 & 1.000 & 3.600 & 0.088 & 14.000 & 0.088 & 0.060 & 8.800 & 11.400 & 3.600 & 0.133 & 11.400 & 11.400 & 0.114 \\
0.065 & 3.600 & 0.278 & 1.000 & 0.071 & 11.400 & 0.071 & 0.052 & 6.200 & 8.800 & 1.000 & 0.099 & 8.800 & 8.800 & 0.088 \\
0.435 & 16.600 & 11.400 & 14.000 & 1.000 & 24.400 & 1.000 & 0.161 & 19.200 & 21.800 & 14.000 & 4.900 & 21.800 & 21.800 & 3.600 \\
0.435 & 16.600 & 11.400 & 14.000 & 1.000 & 24.400 & 1.000 & 0.161 & 19.200 & 21.800 & 14.000 & 4.900 & 21.800 & 21.800 & 3.600 \\
0.049 & 0.278 & 0.114 & 0.161 & 0.052 & 6.200 & 0.052 & 0.041 & 1.000 & 3.600 & 0.161 & 0.065 & 3.600 & 3.600 & 0.060 \\
0.043 & 0.161 & 0.088 & 0.114 & 0.046 & 3.600 & 0.046 & 0.037 & 0.278 & 1.000 & 0.114 & 0.056 & 1.000 & 1.000 & 0.052 \\
0.065 & 3.600 & 0.278 & 1.000 & 0.071 & 11.400 & 0.071 & 0.052 & 6.200 & 8.800 & 1.000 & 0.099 & 8.800 & 8.800 & 0.088 \\
0.161 & 12.700 & 7.500 & 10.100 & 0.204 & 20.500 & 0.204 & 0.099 & 15.300 & 17.900 & 10.100 & 1.000 & 17.900 & 17.900 & 0.435 \\
0.043 & 0.161 & 0.088 & 0.114 & 0.046 & 3.600 & 0.046 & 0.037 & 0.278 & 1.000 & 0.114 & 0.056 & 1.000 & 1.000 & 0.052 \\
0.043 & 0.161 & 0.088 & 0.114 & 0.046 & 3.600 & 0.046 & 0.037 & 0.278 & 1.000 & 0.114 & 0.056 & 1.000 & 1.000 & 0.052 \\
0.204 & 14.000 & 8.800 & 11.400 & 0.278 & 21.800 & 0.278 & 0.114 & 16.600 & 19.200 & 11.400 & 2.300 & 19.200 & 19.200 & 1.000
\end{bmatrix}
\tag{21}
$$

Then, based on the judgment matrix $B$ (see Equation (21)), Equations (7) and (8) are used to construct a quasi-optimal consistent matrix $D$, as shown in the matrix (22).

$$
\begin{bmatrix}
93.413 & 5.575 & 14.478 & 9.068 & 71.883 & 1.265 & 71.883 & 158.573 & 3.525 & 2.075 & 9.068 & 35.239 & 2.075 & 2.075 & 44.525 \\
0.745 & 0.044 & 0.115 & 0.072 & 0.573 & 0.010 & 0.573 & 1.265 & 0.028 & 0.017 & 0.072 & 0.281 & 0.017 & 0.017 & 0.355 \\
55.029 & 3.284 & 8.529 & 5.342 & 42.346 & 0.745 & 42.346 & 93.413 & 2.076 & 1.222 & 5.342 & 20.759 & 1.222 & 1.222 & 26.229 \\
3.284 & 0.196 & 0.509 & 0.319 & 2.527 & 0.044 & 2.527 & 5.575 & 0.124 & 0.073 & 0.319 & 1.239 & 0.073 & 0.073 & 1.565 \\
8.529 & 0.509 & 1.322 & 0.828 & 6.563 & 0.115 & 6.563 & 14.478 & 0.322 & 0.189 & 0.828 & 3.217 & 0.189 & 0.189 & 4.065 \\
5.342 & 0.319 & 0.828 & 0.519 & 4.111 & 0.072 & 4.111 & 9.068 & 0.202 & 0.119 & 0.519 & 2.015 & 0.119 & 0.119 & 2.546 \\
42.346 & 2.527 & 6.563 & 4.111 & 32.586 & 0.573 & 32.586 & 71.883 & 1.598 & 0.941 & 4.111 & 15.974 & 0.941 & 0.941 & 20.184 \\
42.346 & 2.527 & 6.563 & 4.111 & 32.586 & 0.573 & 32.586 & 71.883 & 1.598 & 0.941 & 4.111 & 15.974 & 0.941 & 0.941 & 20.184 \\
2.076 & 0.124 & 0.322 & 0.202 & 1.598 & 0.028 & 1.598 & 3.5258 & 0.078 & 0.046 & 0.202 & 0.783 & 0.046 & 0.046 & 0.990 \\
1.222 & 0.073 & 0.189 & 0.119 & 0.941 & 0.017 & 0.941 & 2.075 & 0.046 & 0.027 & 0.119 & 0.461 & 0.027 & 0.027 & 0.583 \\
5.342 & 0.319 & 0.828 & 0.519 & 4.111 & 0.072 & 4.111 & 9.068 & 0.202 & 0.119 & 0.519 & 2.015 & 0.119 & 0.119 & 2.546 \\
20.759 & 1.239 & 3.217 & 2.015 & 15.974 & 0.281 & 15.974 & 35.239 & 0.783 & 0.461 & 2.015 & 7.831 & 0.461 & 0.461 & 9.895 \\
1.222 & 0.073 & 0.189 & 0.119 & 0.941 & 0.017 & 0.941 & 2.075 & 0.046 & 0.027 & 0.119 & 0.461 & 0.027 & 0.027 & 0.538 \\
1.222 & 0.073 & 0.189 & 0.119 & 0.941 & 0.017 & 0.941 & 2.075 & 0.046 & 0.027 & 0.119 & 0.461 & 0.027 & 0.027 & 0.538 \\
26.229 & 1.565 & 4.065 & 2.546 & 20.184 & 0.355 & 20.184 & 44.525 & 0.990 & 0.583 & 2.546 & 9.895 & 0.583 & 0.583 & 12.502
\end{bmatrix}
\tag{22}
$$

Finally, the subjective weight $W1$ is determined according to Equations (9) and (10); the results are shown in the matrix (23).

$$
\begin{bmatrix}
0.302 & 0.002 & 0.178 & 0.011 & 0.028 & 0.017 & 0.137 & 0.137 & 0.007 & 0.004 & 0.017 & 0.067 & 0.004 & 0.004 & 0.085
\end{bmatrix}^{T}
\tag{23}
$$

2.  Objective Weight calculation

Based on the evaluation index parameters (see Table 5), Equation (11) is adopted to construct the decision matrix $X$. The decision matrix $X$ is shown in the matrix (24).

$$
\begin{bmatrix}
45 & 13 & 9.1 & 0.25 & 12 & 37 & 19.7 & 0.0405 & 15.0 & 6.2 & 460 & \text{High} & \text{Good} & \text{Good} & \text{High} \\
45 & 20 & 8.3 & 0.28 & 25 & 35 & 16.2 & 0.0035 & 15.5 & 5.5 & 450 & \text{High} & \text{Good} & \text{Good} & \text{High} \\
45 & 16 & 9.0 & 0.23 & 10 & 38 & 19.7 & 0.0347 & 15.0 & 6.0 & 460 & \text{High} & \text{Good} & \text{Good} & \text{High}
\end{bmatrix}
\tag{24}
$$

Secondly, using Equations (12)–(14) and the decision matrix $X$ (see Equation (23)), entropy vector $p$ is calculated, as shown in the matrix (25).

$$
\begin{bmatrix}
0.050 & 0.049 & 0.049 & 0.068 & 0.091 & 0.065 & 0.097 & 0.097 & 0.097 & 0.097 & 0.049 & 0.049 & 0.049 & 0.049 & 0.049 \\
0.050 & 0.100 & 0.100 & 0.100 & 0.050 & 0.100 & 0.050 & 0.050 & 0.050 & 0.050 & 0.100 & 0.050 & 0.050 & 0.050 & 0.050 \\
0.050 & 0.071 & 0.056 & 0.050 & 0.099 & 0.050 & 0.099 & 0.092 & 0.099 & 0.085 & 0.050 & 0.050 & 0.050 & 0.050 & 0.050
\end{bmatrix}
\tag{25}
$$

Finally, Equation (15) is employed to calculate the objective weight $W_2$. The results are shown in the matrix (26).

$$\begin{bmatrix} 0.077 & 0.063 & 0.066 & 0.063 & 0.059 & 0.064 & 0.058 & 0.058 & 0.058 & 0.060 & 0.067 & 0.077 & 0.077 & 0.077 & 0.077 \end{bmatrix}^T \quad (26)$$

### 4.2.3. Combined Weight Calculation

Firstly, according to Equation (16), the distance function can be obtained $d(w_1, w_2) = 0.323$, which passes the consistency test of combination weighting in the interval $[0, 1]$.

Then, using Equation (17), the weight coefficient is calculated as

$$\begin{cases} p = 0.661 \\ q = 0.339 \end{cases} \quad (27)$$

Finally, Equation (18) is employed to determine the total weight $w$. The results are shown in the matrix (28).

$$\begin{bmatrix} 0.153 & 0.042 & 0.104 & 0.045 & 0.048 & 0.048 & 0.085 & 0.085 & 0.041 & 0.041 & 0.050 & 0.073 & 0.052 & 0.052 & 0.079 \end{bmatrix}^T \quad (28)$$

The evaluation index weight calculation results in Sections 4.2.2 and 4.2.3 are summarized in the weight of evaluation indexes shown in Table 8.

**Table 8.** Weights of evaluation indexes.

| Index | Subjective Weight | Objective Weight | Comprehensive Weight |
|---|---|---|---|
| $C_{11}$ | 0.302 | 0.077 | 0.153 |
| $C_{12}$ | 0.002 | 0.063 | 0.042 |
| $C_{21}$ | 0.178 | 0.066 | 0.104 |
| $C_{22}$ | 0.011 | 0.063 | 0.045 |
| $C_{23}$ | 0.028 | 0.059 | 0.048 |
| $C_{24}$ | 0.017 | 0.064 | 0.048 |
| $C_{25}$ | 0.137 | 0.058 | 0.085 |
| $C_{26}$ | 0.137 | 0.058 | 0.085 |
| $C_{31}$ | 0.007 | 0.058 | 0.041 |
| $C_{32}$ | 0.004 | 0.060 | 0.041 |
| $C_{33}$ | 0.017 | 0.067 | 0.050 |
| $C_{41}$ | 0.067 | 0.077 | 0.073 |
| $C_{42}$ | 0.004 | 0.077 | 0.052 |
| $C_{43}$ | 0.004 | 0.077 | 0.052 |
| $C_{44}$ | 0.085 | 0.077 | 0.079 |

### 4.2.4. Calculation of Comprehensive Connection Degree

Tables 7 and 8 calculate the comprehensive connection degree of each construction section using Equation (19), and the results are shown in Equation (29).

$$\begin{aligned} \mu_1 &= 0.193 + 0.312i_1 + 0.201i_2 + 0.167i_3 + 0.127j \\ \mu_2 &= 0.196 + 0.283i_1 + 0.225i_2 + 0.255i_3 + 0.041j \\ \mu_3 &= 0.211 + 0.101i_1 + 0.333i_2 + 0.238i_3 + 0.117j \end{aligned} \quad (29)$$

### 4.3. Results and Discussion

#### 4.3.1. Evaluation Result Analysis

The value $i$ is taken according to the principle of equal division, so the eigenvector matrix $E = (1, 0.5, 0, -0.5, -1)^T$ is substituted into Equation (29) to calculate the comprehensive connection degree of each evaluation object. The safety evaluation results are shown in Table 9.

**Table 9.** Safety evaluation result.

| Object of Evaluation | Comprehensive Connection Degree | Safety Level | Risk Level |
| --- | --- | --- | --- |
| $L_1$ | 0.139 | III | medium level |
| $L_2$ | 0.168 | III | medium level |
| $L_3$ | 0.025 | III | medium level |

According to the construction safety evaluation model of subway tunnels under extreme weather conditions based on the multivariate linkage number and set pair analysis theory, the comprehensive linkage degrees of three construction sections, $L_1$, $L_2$, and $L_3$, of the city rail Line 2 tunnel are 0.139, 0.168, and 0.025, respectively, all within the interval $(-0.2, 0.2]$; therefore, all of them are at safety level III. Therefore, the overall safety level of the subway tunnel in this area is a medium level, which is a general risk and is consistent with the actual survey situation.

Construction sections $L_1$, $L_2$, and $L_3$ are a general risk. The comprehensive connection degree of construction section $L_1$ is $\mu_1 = 0.193 + 0.312i_1 + 0.201i_2 + 0.167i_3 + 0.127j$, where the same degree $a = 0.193$ is greater than the opposite degree, $c = 0.127$, and the different degree $b = (0.312, 0.201, 0.167)$ is greater than the same degree, $a = 0.193$. The comprehensive connection degree of construction section $L_2$ is $\mu_2 = 0.196 + 0.283i_1 + 0.225i_2 + 0.255i_3 + 0.041j$, where the same degree $a = 0.196$ is greater than the opposite degree, $c = 0.041$, and the different degree $b = (0.283, 0.225, 0.255)$ is greater than the same degree $a = 0.174$. The comprehensive connection degree of construction section $L_3$ is $\mu_3 = 0.211 + 0.101i_1 + 0.333i_2 + 0.238i_3 + 0.117j$, where the same degree $a = 0.211$ is greater than the opposite degree, $c = 0.117$, and the different degree $b = (0.101, 0.333, 0.238)$ is greater than the same degree $a = 0.211$. As shown in Table 2, construction sections $L_1$, $L_2$, and $L_3$ all belong to the micro-identity, with weak homogeneity and weak improvement trend, indicating that construction sections $L_1$, $L_2$, and $L_3$ are not likely to develop in the direction of "lower risk." The evaluation results are consistent with the actual survey, which indicates that the evaluation model is feasible and universal.

### 4.3.2. Discussion

We used the IAHP and EWM to calculate the subjective and objective weights of the indexes, respectively, introducing the idea of linear weighting to test the evaluation index weights' consistency and combine them. This can not only avoid the excessive interference of human factors but also eliminate the excessive proportion of objective factors, giving full play to the advantages of each weighting method and overcoming the limitations caused by using a single method. In addition, the combination weight based on the linear weighting method can always find a balance between the majority of similar and a few different results. More moderate and explanatory weights were obtained than methods used in other studies, and relevant evaluation results were obtained [20,30,44,53]. Therefore, this method is a more comprehensive and systematic weight determination method.

It can be observed from Table 8 that the subjective and objective weights of the 15 indicators have little deviation and correspond to each other. As observed in other studies, hydrogeological and management factors' impact on subway tunnels' construction safety under extreme rainfall weather is significant [4,11].

The subjective weights of rainfall ($C_{11}$), groundwater level ($C_{21}$), water content ($C_{25}$), underground permeability coefficient ($C_{26}$), monitoring and detection strength ($C_{41}$), and professional skills of the construction personnel ($C_{44}$) are all higher than the average index weight of 0.066, showing that experts and decision makers subjectively believe that the six evaluation above indicators are the most important. From the results of the EWM model's calculations, it is clear that the objective weights of the seven evaluation indicators, namely, rainfall ($C_{11}$), groundwater level ($C_{21}$), lining thickness ($C_{33}$), monitoring and detection intensity (C41), construction organization design ($C_{42}$), safety organization and system ($C_{43}$), and professional skills of the construction personnel ($C_{44}$), are all greater than

the average index weight of 0.066, indicating that, from the perspective of the evaluation indicators, the design of the construction organization and from the calculation results of the linear weighting method, it can be seen that the total weights of rainfall ($C_{11}$), groundwater level ($C_{21}$), water content ($C_{25}$), underground permeability coefficient ($C_{26}$), tunnel depth ($C_{31}$), monitoring and detection strength ($C_{41}$), and professional skills of the construction personnel ($C_{44}$) are more significant than the average index weight of 0.066, indicating that these are the main factors affecting the safety of subway tunnel construction under extreme rainfall weather conditions.

Based on the above, both the subjective and objective weights and the total weights of the indicators (rainfall ($C_{11}$), underground water level ($C_{21}$), water content ($C_{25}$), underground permeability coefficient ($C_{26}$), lining thickness ($C_{33}$), monitoring and detection strength ($C_{41}$), and professional skills of the construction personnel ($C_{44}$)) have significant weights, showing that these are the main factors that affect the safety of subway tunnel construction under extreme conditions. Although the other eight evaluation indexes have a lesser impact on the safety of subway tunnel construction than the above seven evaluation indexes, they also pose a particular threat to the safety of subway tunnels under extreme rainfall weather conditions with the changes in time and space.

In contrast to traditional evaluation methods [8,28,31,39], we introduce the multivariate communication number set pair analysis theory to build a combination weighting–set pair analysis safety evaluation model. We quantitatively calculated the subway tunnel construction safety level under extreme rainfall weather conditions. Moreover, we analyzed the risk development trend while also solving the fuzziness and uncertainty of the evaluation index and co-efficient. During tunnel construction, if it is found that the risk level is high and does not meet the risk acceptance criteria, it is necessary to adjust the main risk factors dynamically. Then, using the method proposed in this study, it is easier to obtain the adjusted safety level of subway tunnel construction. Therefore, compared with the traditional evaluation methods [27,32,33,54,55], this methodology is more practical and scientific in forecasting and reducing the incidence of subway tunnel construction safety incidents.

## 5. Conclusions

(1) We studied the safety risk of subway tunnel construction under heavy rainfall conditions. The damaging effects of heavy rainfall on the subway tunnel structural system and the influence factors of internal construction status on subway tunnel construction safety were identified.

(2) Evaluating subway tunnel safety during heavy rains is a complicated, methodical, multidisciplinary issue. In this study, the central safety evaluation index system and its assessment standards were developed based on a complete investigation of the safety risk elements for subway tunnel construction during extreme rainfall, including disaster-inducing variables and disaster-nurturing environments.

(3) Based on the idea of linear weighting, we optimized the combination of subjective and objective weights calculated by IAHP and EWM. More importantly, the method considers the subjective values of decision makers and the objectivity contained in the data, which makes the calculation results of the study closer to reality. Then, to address the ambiguity and uncertainty of the evaluation indexes, we introduced the SPA theory and made a quantitative judgment on the safety level and development trend of subway tunnel construction under extreme weather conditions by establishing pooled pairs.

(4) Identifying the influencing factors on the safety of subway tunnel construction under extreme rainfall weather conditions in this study is not comprehensive, and there are various uncertainties. In reality, identifying such factors and the solution methods are still to be improved. In addition, sensitivity analysis of these influencing factors is needed in future studies.

**Author Contributions:** Methodology, Y.W.; investigation, J.X.; resources, Q.L.; data curation, Y.W.; writing—original draft preparation, Y.W.; writing—review and editing, T.H.; supervision, P.L.; project administration, T.H., P.L.; funding acquisition, P.L. All authors have read and agreed to the published version of the manuscript.

**Funding:** This research was funded by the National Natural Science Foundation of China (grant number 72061023), the Natural Science Foundation of Gansu Province (grant number 20JR10RA173), and the Hongliu Outstanding Young Talents Support Program of Lanzhou University of Technology (grant number 0320038).

**Institutional Review Board Statement:** Not applicable.

**Informed Consent Statement:** Not applicable.

**Data Availability Statement:** Not applicable.

**Acknowledgments:** The authors' special thanks go to all experts and survey participants of the paper. This work was supported by the National Natural Science Foundation of China, the Natural Science Foundation of Gansu Province, China, and the Hongliu Outstanding Young Talents Support Program of Lanzhou University of Technology.

**Conflicts of Interest:** The authors declare that they have no known competing financial interests or personal relationships that could have appeared to influence the work reported in this paper.

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
