# Peer review of "Safety Evaluation of Subway Tunnel Construction under Extreme Rainfall Weather Conditions Based on Combination Weighting–Set Pair Analysis Model"

_sustainability, doi:10.3390/su14169886_

Round 1

Reviewer 1 Report

The text is very well explained from a mathematical point of view, however the text is confusing, making it difficult to understand the tables and equations.

Based on this, it would be necessary to better describe where the contributions of the work are, whether it is in the modeling or just in the application of it to the proposed problem. Based on this, it will be easier to understand the scientific merit of the work. Are there similar applications of the models proposed in the literature to justify their choice?

Author Response

We wish to thank all the Reviewers and Editors for their thoughtful comments that help us improve this paper. In this revision, we corrected the errors and clarified the points and concerns addressed by the Reviewers. Please refer to the following responses for the details about how we take into all Reviewers’ comments in preparing this revision.

Reviewer #1:

The text is very well explained from a mathematical point of view, however the text is confusing, making it difficult to understand the tables and equations.

Comment No.1: (The text is very well explained from a mathematical point of view however, the text is confusing, making it difficult to understand the tables and equations.)

Response comment No.1: Thanks for the reviewers’ approval of this study. Based on your valuable comments, we have further explained some of the tables in this paper and simplified the complex statements—for example, Table 7, 8 and equations (19)-(28).

Comment No.2: (Based on this, it would be necessary to better describe where the contributions of the work are, whether it is in the modeling or just in the application of it to the proposed problem. Based on this, it will be easier to understand the scientific merit of the work. Are there similar applications of the models proposed in the literature to justify their choice?)

Response comment No.2: Considering the reviewer’s suggestion, We further describe the contribution of this study to modeling and engineering applications in the Abstract, end of the Introduction section and the Conclusion section. The details are shown below:

“To overcome the deficiencies mentioned above, this work develops a safety evaluation model based on the multivariate linkage number and set pair analysis theory, considering extreme precipitation conditions. The model uses the benefits of set pair analysis to look at the system's deterministic and uncertain problems. Not only does it take into account how vague and uncertain the indicators are, but it also fixes problems with the traditional evaluation model, in which the values of the indicators are seen as fixed. The linear weighting idea is also introduced to perform the optimal combination of the indicator weights calculated by the improved hierarchical analysis and entropy weight method, which makes the weighting calculation more scientific and the evaluation results more accurate.”

“(1)This article studies the safety risk of subway-tunnel construction under heavy rainfall conditions. The damaging effects of heavy rainfall on the subway tunnel structural system and the influence factors of internal construction status on subway tunnel construction safety are identified.

(2)Evaluating subway tunnel safety during heavy rains is a complicated, methodical, multidisciplinary issue. In this study, the central safety evaluation index system and its assessment standards were developed based on a complete investigation of the safety risk elements for subway tunnel construction during extreme rainfall, including disaster inducing variables and disaster-breeding environments.

(3) Based on the idea of linear weighting, this study optimizes the combination of subjective and objective weights calculated by IAHP and EWM. More importantly, the method considers the subjective values of decision makers and the objectivity contained in the data, which makes the calculation results of the study closer to reality. Then, to address the ambiguity and uncertainty of the evaluation indexes, this study introduces the SPA theory and makes a quantitative judgment on the safety level and development trend of subway tunnel construction under extreme weather conditions by establishing pooled pairs.”

The evaluation model used in this study is an innovative point of this study, and the theoretical rationality of the method used is introduced accordingly in the Method section of the article, and the relevant literature has been cited as proof. The details are shown below:

“SPA has been widely used in many fields, such as tunnel collapse, tailings reservoir, and port ecological risk assessment. Using SPA, Chen et al. [43] provided a thorough evaluation technique of mountain tunnel collapse risk. A SPA quantitative risk assessment technique built on a fuzzy evaluation method was proposed by Shi et al. [44]. Li et al. [45]construction of the port ecological assessment model utilized the SPA theory.

Reviewer 2 Report

This paper presents a model for the evaluation of the construction safety of subway tunnels under extreme rainfall conditions. I think the structure of the article. There are a few questions and suggestions, though, that I think might be helpful for the authors to consider for suggested minor revision prior to the publication of the manuscript:

1.      Table 4: Can the authors explain that why ‘safe’ is written twice in the Grading Standard.

2.      Table 4 & 6: There are some values in the table such as the values of C41-C44, which are not quantified. How were these values used in the proposed model?

3.      Figure 3: The figure is not clear. Please use a high resolution picture.

Author Response

We wish to thank all the Reviewers and Editors for their thoughtful comments that help us improve this paper. In this revision, we corrected the errors and clarified the points and concerns addressed by the Reviewers. Please refer to the following responses for the details about how we take into all Reviewers’ comments in preparing this revision.

Reviewer #2:

This paper presents a model for the evaluation of the construction safety of subway tunnels under extreme rainfall conditions. I think the structure of the article. There are a few questions and suggestions, though, that I think might be helpful for the authors to consider for suggested minor revision prior to the publication of the manuscript:

Comment No.1: (Table 4: Can the authors explain that why ‘safe’ is written twice in the Grading Standard.)

Response comment No.1: First of all, thanks for the reviewers’ approval of this study. Due to errors in language editing, "basically safety" in Table 4 was incorrectly edited as "safety," which has been corrected.

Comment No.2: (Table 4 & 6: There are some values in the table such as the values of C41-C44, which are not quantified. How were these values used in the proposed model?)

Response comment No.2: The authors appreciate the reviewers’ constructive suggestions. Considering the reviewer’s suggestions, we have added the quantification rules for indicators C41-C44 in Table 4. and added the definition of quantification rules in the first paragraph of Section 3.3 of Part “Safety evaluation model”.

Comment No.3: (3. Figure 3: The figure is not clear. Please use a high resolution picture.)

Response to comment No.3: Thanks for the reviewers’ valuable comments. Based on the reviewer’s comments, we have replaced Figure 3 with a more precise image.

Reviewer 3 Report

The article concerns research on safety evaluation of subway tunnel construction under extreme rainfall weather conditions based on combination weighting-set pair analysis model.

 The article should be edited because it does not meet the requirements of the journal of Sustainability (e.g. different font sizes in equations, additional lines in tables, tables divided on several pages - you can rearrange the text so that the table is on one page).

 The paper should be improved namely in some identified aspects:

- the authors should have some extra effort to make the description more easily understandable, more clear to all the readers and more simple to read as the subject seems a kind of too “complicated”.

- figure 3 is not clear enough.           

- although the English is OK, the text needs to be simplified.

- the paper omits the basic phenomenon as it is principal stress rotation. This phenomenon is described among others in the following papers: (1) Anisotropy of soil shear strength parameters caused by the principal stress rotation - journal Archives of Civil Engineering; (2) Evaluation of the Change in Undrained Shear Strength in Cohesive Soils due to Principal Stress Rotation Using an Artificial Neural Network - Applied Sciences.

- there should be more references to existing scientific achievements in the article, because the literature review omits the basic achievements in recent years (examples of achievements were published, among others, in the journal Applied Sciences, Sustainability).

Author Response

We wish to thank all the Reviewers and Editors for their thoughtful comments that help us improve this paper. In this revision, we corrected the errors and clarified the points and concerns addressed by the Reviewers. Please refer to the following responses for the details about how we take into all Reviewers’ comments in preparing this revision.

Reviewer #3:

The article concerns research on safety evaluation of subway tunnel construction under extreme rainfall weather conditions based on combination weighting-set pair analysis model.

Comment No.1: (The article should be edited because it does not meet the requirements of the journal of Sustainability (e.g. different font sizes in equations, additional lines in tables, tables divided on several pages - you can rearrange the text so that the table is on one page.)

Response comment No.1: The authors appreciate the reviewers’ constructive suggestions. Based on reviewers’ valuable comments, we have revised some of the tables based on the editorial layout of the Sustainability Journal, as shown in Table 7.

Comment No.2: (the authors should have some extra effort to make the description more easily understandable, more clear to all the readers and more simple to read as the subject seems a kind of too “complicated”)

Response comment No.2: The authors appreciate the reviewers’ careful guidance. Based on reviewers' comments, we asked the English experts to simplify some sentences, Tables and equations in the article to make them more readable. For example, Table7 and equation (19-28).

Comment No.3: (figure 3 is not clear enough.)

Response to comment No.3: Thanks for the reviewers’ valuable comments. Based on your comments, we have replaced Figure 3 with a more precise image.

Comment No.4: (the paper omits the basic phenomenon as it is principal stress rotation. This phenomenon is described among others in the following papers: (1) Anisotropy of soil shear strength parameters caused by the principal stress rotation - journal Archives of Civil Engineering; (2) Evaluation of the Change in Undrained

Shear Strength in Cohesive Soils due to Principal Stress Rotation Using an Artificial Neural Network -Applied Sciences.)

Response to comment No.4: Thanks for the reviewers’ valuable comments on this article. Based on the reviewers’ comments, we have improved the literature review section and cited the reviewers’ recommended article at the end of the third paragraph. The details are shown below:

“Grzegorz Wrzesi ´nski et al.[14-15] applied artificial neural networks to evaluate the varia-tion of undrained shear strength in cohesive soils due to rotation of principal stresses.”

Comment No.5: (there should be more references to existing scientific achievements in the article, because the literature review omits the basic achievements in recent years (examples of achievements were published, among others, in the journal Applied Sciences, Sustainability.)

Response to comment No.5: Thanks for the reviewers’ valuable comments on this article. Based on the reviewer's comments, we have improved the literature review section at the end of the third paragraph. The details are shown below: 

“Sang-Guk Yum et al.[16] developed a new tunnel-centered natural disaster risk assess-ment method by performing multiple linear regression analyses on financial loss data generated from tunnel construction in Korea. Jianxiu Wang et al.[17] used AHP to determine the baseline weights of the tunnel construction dynamic risk assessment indexes. Ping Liu et al.[18] have proposed a support vector machine model based on a particle swarm algorithm for forecasting the safety risks in metro construction.”

This manuscript is a resubmission of an earlier submission. The following is a list of the peer review reports and author responses from that submission.